# How Does Layer Normalization Improve Deep $Q$-learning?

## Abstract

Layer normalization (LN) is among the most effective normalization schemes for deep $Q$-learning. However, its benefits remain not fully understood. We study LN through the lens of *gradient interference*. A gradient interference metric used in prior works is the inner product between semi-gradients of the temporal difference error on two random samples. We argue that, from the perspective of minimizing the loss, a more principled metric is to calculate the inner product between a semi-gradient and a full-gradient. We test this argument with offline deep $Q$-learning, without a target network, on four classic control tasks. However, counterintuitively, we find empirically that first-order gradient interference metrics *positively* correlate with the training loss. We empirically show that adding a second-order gradient interference term gives more intuitive results. Theoretically, we provide supporting arguments from the linear regression setting.

## 1 Introduction

Deep $Q$-learning, including deep $Q$-networks (DQN) (Mnih et al., 2013), is an important reinforcement learning (RL) method with wide applications including robotics (Gu et al., 2017), autonomous driving (Sallab et al., 2017), and healthcare (Yu et al., 2021). A key characteristic of DQN is that it is an off-policy algorithm that directly learns the optimal value function, which potentially improves sample efficiency and reduces the risks associated with on-policy exploration.

However, the benefit of off-policy learning often comes with optimization instabilities. DQN is prone to error propagation across iterations which could lead to performance drop or gradient explosion (Luo et al., 2024). Many techniques have been developed to help stabilize DQN, including target networks (Riedmiller, 2005; Mnih et al., 2015), replay buffers (Lin, 1992; Mnih et al., 2013), prioritized replay (Schaul et al., 2015), and double $Q$-learning (Van Hasselt, 2010; Van Hasselt et al., 2016).

One particularly simple yet effective stabilization technique is Layer Normalization (LN). Empirically, its benefits can rival — or even surpass — those of the RL-specific techniques above. For example, several works have empirically shown that LN can somewhat replace the benefits of a target network. Interestingly, related methods successful in supervised learning, such as batch normalization and weight normalization, do not typically provide similar benefits in RL (*cf.* Bhatt et al. (2024)). This raises the question of why LN, in particular, has such a strong effect on DQN's performance without additional tricks. Notably, this discrepancy between LN and other normalization methods also extends to LLMs, where only LN has proven effective (Shen et al., 2020).

Table 1: Mean return of 20 seeds, each averaging over all gradient steps of 9 runs, one run for each of 9 learning rates. The highest return per environment is highlighted, as well as the return for environments whose 95% CI overlaps the CI of the highest mean. "None" means no normalization is used. Trained with SGD. On Pendulum, random actions outperform all algorithms other than LN. Details in Appendix G.

| | Random | None | LN | BN | WN |
|---|---|---|---|---|---|
| **Acrobot** | -452.604 | -384.0 | -267.2 | -477.6 | -409.6 |
| **Pendulum** | -1281.860 | -1323.2 | -792.8 | -1433.7 | -1376.5 |
| **CartPole** | 19.545 | 163.5 | 289.0 | 19.0 | 270.0 |
| **MountainCar** | -194.632 | -163.8 | -122.2 | -169.0 | -166.6 |

Many works have reported the empirical benefits of LN in RL, but few try to provide theoretical explanations. An exception is Gallici et al. (2025), which offers theoretical analysis and empirical backing for deep $Q$-learning with LN and $\ell_2$ regularization. However, their theory uses $\ell_2$ regularization, leaving open the understanding of LN *by itself.* Motivated by this aspect, we revisit LN in DQN. We defer a precise comparison between our analysis and theirs to future work.

We ask how LN changes interference in temporal-difference learning. We examine (i) first-order $Q$-value interference ($\mathsf{QGI}_+$), (ii) first-order loss-based metrics using semi-gradients ($\mathsf{SGI}$) and a mixed semi/full-gradient variant ($\mathsf{MGI}$), and (iii) a second-order–corrected metric ($\mathsf{GI}_2$) derived from a Taylor expansion of the TD-loss decrement.

Across four offline control tasks (SGD, no target network), $\mathsf{QGI}_+$ and $\mathsf{SGI}/\mathsf{MGI}$ *positively* correlate with training loss, whereas $\mathsf{GI}_2$ *negatively* correlates with it; LN reduces first-order interference, increases $\mathsf{GI}_2$, and improves return. We support this with a simple linear-regression argument showing that feature normalization improves an SGD progress–stability tradeoff, and we provide complementary evidence on isometry and implicit learning-rate decay.

It turns out finding a clean explanation that shows a strict improvement from LN is challenging. Given the complexity of DQN — combining deep learning, semi-gradients, and off-policy training — we believe that LN's effect is likely to be multi-faceted. In this paper, we take a step toward understanding its effect from multiple perspectives.

For each perspective, we provide theoretical hypotheses and empirical evidence. While no single perspective gives a full explanation, together they provide a big picture of why LN stabilizes and accelerates DQN. We hope our first step will inspire future study that advances the theory of both RL and deep learning.

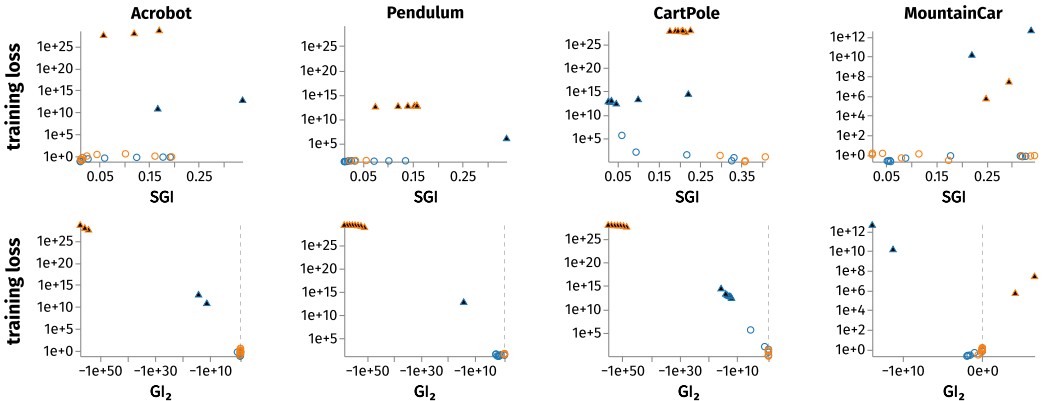

Figure 1: Deep $Q$-learning with LN ⊙ or no normalization ⊙. A triangle ▲ indicates at least one seed had its network weights clamped to avoid NaNs (Appendix G). **First row:** On four classic control tasks, we counterintuitively find that first-order gradient interference ($\mathsf{SGI}$, defined later) *positively* correlates with the training loss, despite the standard interpretation of larger gradient interference as better. Moreover, we find LN tends to *decrease* first-order gradient interference, even though it *improves* the training loss. **Second row:** We find that including a second-order gradient interference term gives a metric ($\mathsf{GI}_2$) with a more intuitive (i.e., negative) correlation with the loss. We likewise find that LN tends to increase this metric as it improves the loss and return. MountainCar largely does not follow these trends. **Both rows:** 30 seeds per data point. Trained with SGD. The same plots with return are shown in Appendix I.

## 2 PROBLEM SETTING

$Q$-**networks.** We study offline RL in continuous state spaces with discrete actions, where the state is denoted by $s \in \mathbb{R}^k$ and the action is denoted by $a \in \{1, \dots, m\}$. The reward function is $r(s, a) \in \mathbb{R}$. To approximate the $Q$-function, a standard single-layer neural network parameterizes the $Q$-function as $Q_\theta(s, a) = \langle v_a, (Ws)_+ \rangle$, where $\theta$ denotes the collection of parameters, $W \in \mathbb{R}^{d \times k}$, $v_a \in \mathbb{R}^d$,

and $(x)_+ \in \mathbb{R}^d$ is the ReLU activation function. The parameters $\theta$ are updated via

$$\theta_{t+1} = \theta_t - \eta \bar{\nabla} \ell_{\theta_t}(s_t, a_t, s'_t) \tag{1}$$

where $\eta > 0$ is the learning rate, $\gamma \in [0, 1)$ is the discount factor, $(s_t, a_t, s'_t)$ is drawn from the offline distribution $\mu$, and $\bar{\nabla}\ell_{\theta_t}(s, a, s') \triangleq \left(Q_{\theta_t}(s, a) - r(s, a) - \gamma \max_{a'} Q_{\theta_t}(s', a')\right) \nabla Q_{\theta_t}(s, a)$ is the semi-gradient on the squared TD error of $(s, a, s')$. This method often suffers from instability, which is commonly mitigated by introducing a *target network* (Mnih et al., 2015).

**Layer Normalization.** Throughout this paper, we focus on offline RL, but we speculate that our results apply to online RL with a replay buffer. For experimental and algorithmic simplicity, we use no target network at any point in this paper. We test four discrete-action classic control tasks (Brockman et al., 2016), using an offline dataset of 10,000 uniformly random actions. All returns we report are averaged over all gradient steps over the course of training in order to measure both stability and speed. All our results are averaged over 30 random seeds unless otherwise noted. We primarily study layer normalization,

$$\text{LN}(x) = \frac{x - \bar{x}}{\sqrt{\frac{1}{d}\sum_{i=1}^d (x_i - \bar{x})^2}}, \quad \bar{x} = \frac{1}{d}\sum_{i=1}^d x_i, \tag{2}$$

with which the single-layer $Q$-network is

$$Q_\theta(s, a) = \langle v_a, (\text{LN}(Ws))_+ \rangle. \tag{3}$$

While standard LN includes additional shift and scale parameters (an elementwise affine transform), in our experiments these had little effect on returns, so we all of our results omit them. LN may also be placed after the ReLU rather than before, but we find the placement before about as effective or better. Our experiments further suggest, like prior work, that LN might roughly match or outperform other normalizations, such as batch normalization (BN) (Ioffe & Szegedy, 2015) and weight normalization (WN) (Salimans & Kingma, 2016). Table 1 shows these results.

## 3 How Does Layer Normalization Improve Deep $Q$-learning?

Normalization layers in supervised learning have been explained from many perspectives, including feature orthogonalization (Daneshmand et al., 2021; Meterez et al., 2024) and automatic learning rate tuning (Arora et al., 2018). In this study, we ultimately aim for RL-specific explanations. We make progress on three:

- **Gradient interference**, various metrics[1] for how training gradients between different training samples interact. In this work, these are the metrics we find most compelling for explaining why LN improves deep $Q$-learning (Section 3.1).
- **Isometry**, how clustered the eigenvalues of the Gram matrix of the learned representations are (Joudaki et al., 2023). We derive a new version of this metric for the setting where batches are large compared to feature dimensions, and find LN still improves this new metric in our RL setting (Appendix C). Theoretically connecting this metric to the convergence rate (even in supervised learning) is perhaps the most important open question here.
- **Implicit learning rate decay**, the fact that LN (and other normalizations) implicitly decay the learning rate as the magnitudes of the weights increase throughout training. Aligning with Lyle et al. (2024), our results suggest that implicit learning rate decay might sometimes increase returns, but does not fully account for LN's increase of returns (Appendix D).

### 3.1 Gradient Interference

#### 3.1.1 Motivation from Tabular $Q$-learning

Tabular $Q$-learning provably converges to $Q^\star$ under data coverage and properly tuned learning rates. In tabular $Q$-learning (Watkins et al., 1989), only the $Q$-value of the sampled state-action is *directly* updated. In deep $Q$-learning, every training sample's gradient contributes not only to the direct update

---

[1]Throughout, we use "metric" in the colloquial sense, not in the sense of a distance function.

at that sample, but also *indirect* updates to other state-actions. Such indirect updates are how models *generalize* to unseen state-actions. While generalization is a common goal in machine learning, it can also introduce adverse side effects. For example, in classic examples of $Q$-learning with function approximation (Baird's and $w$-to-$2w$ examples (Sutton et al., 1998)), the model diverges due to the undesired correlation between the $Q$-value of indirectly updated $Q$-values and those of directly updated $Q$-values. This becomes an issue in $Q$-learning but not in supervised learning as $Q$-learning is not based on gradient descent.

We refer to such harmful generalization as "interference", which, as those classic works show, plays a more central role in RL than in supervised learning. We remark that although a lot of RL research studies out-of-distribution generalization (e.g., to new state distributions induced by new policies), our focus here is on in-distribution interference, which primarily affects the stability of the learning dynamics rather than generalization across distributions. To eliminate the confounding effect of out-of-distribution states, we generate training data in all our experiments from a broad state distribution. We focus on this in-distribution generalization.

Below, we compare in more detail the updates of tabular $Q$-learning (which lacks generalization and interference, and provably converges under mild conditions) and deep $Q$-learning (with generalization and interference, and provably diverges in some simple cases). Recall the tabular $Q$-learning algorithm (Watkins et al., 1989):

$$Q_{t+1}(s, a) = Q_t(s, a) - \eta \Delta_t(s_t, a_t, s_t') \mathbb{I}[(s, a) = (s_t, a_t)], \tag{4}$$

where $\Delta_t(s, a, s') = Q_t(s, a) - r(s, a) - \gamma \max_{a'} Q_t(s', a')$ is the (unsquared) TD error of $(s, a, s')$, and $\mathbb{I}[\cdot]$ is the indicator function.

In deep $Q$-learning, we have by first-order approximation and Eq. (1):

$$\begin{aligned} Q_{\theta_{t+1}}(s, a) &\approx Q_{\theta_t}(s, a) + \langle \theta_{t+1} - \theta_t, \nabla Q_{\theta_t}(s, a) \rangle \\ &= Q_{\theta_t}(s, a) - \eta \left\langle \nabla \ell_{\theta_t}(s_t, a_t, s_t'), \nabla Q_{\theta_t}(s, a) \right\rangle \\ &= Q_{\theta_t}(s, a) - \eta \Delta_{\theta_t}(s_t, a_t, s_t') \left\langle \nabla Q_{\theta_t}(s_t, a_t), \nabla Q_{\theta_t}(s, a) \right\rangle. \end{aligned} \tag{5}$$

Comparing tabular $Q$-learning (Eq. (4)) and deep $Q$-learning (Eq. (5)), we observe that **(1)** for the directly updated state-action $(s_t, a_t)$, the changes of its $Q$-value in the two algorithms are related by $Q_{\theta_{t+1}}(s_t, a_t) - Q_{\theta_t}(s_t, a_t) = (Q_{t+1}(s_t, a_t) - Q_t(s_t, a_t)) \|\nabla Q_{\theta_t}(s_t, a_t)\|^2$, meaning that the two algorithms update $Q(s_t, a_t)$ in the same direction. We also observe **(2)** for other state-actions $(s, a) \neq (s_t, a_t)$, tabular $Q$-learning keeps $Q(s, a)$ unchanged, while deep $Q$-learning modifies $Q_{\theta_{t+1}}(s, a) = Q_{\theta_t}(s, a) - \eta \Delta_t(s_t, a_t, s_t') \langle \nabla Q_{\theta_t}(s_t, a_t), \nabla Q_{\theta_t}(s, a) \rangle$ via indirect updates. In other words, in deep $Q$-learning, direct updates may *interfere* with the $Q$-values of other state-actions, through indirect updates. Using the absolute cosine similarity notation $\cos_+(x, y) \triangleq \frac{|\langle x, y \rangle|}{\|x\|\|y\|}$, one way to quantify this interference is the absolute **$Q$-value gradient interference**,

$$\mathsf{QGI}_+ = \mathbb{E}_{(s,a),(s_t,a_t) \sim \mu \mid (s,a) \neq (s_t,a_t)} \left[ \cos_+ \left( \nabla Q_{\theta_t}(s_t, a_t), \nabla Q_{\theta_t}(s, a) \right) \right]. \tag{6}$$

Using the absolute value reflects our aim with $\mathsf{QGI}_+$ to measure all interference in any direction. Zero-mean interference would still behave far differently than tabular $Q$-learning if the variance over training samples was high. Using the cosine similarity avoids confounding the average directions of the interference with the magnitudes of the gradients, though see also the later Section 3.3 for another approach to avoid such confounding.

$\mathsf{QGI}_+$ **positively correlates with training loss in our experiments.** These $\mathsf{QGI}_+$ results (row two of Fig. 2) somewhat align with our hypothesis that larger interference magnitude might cause instability, increasing loss and decreasing return. However, one limitation of the interference metric $\mathsf{QGI}_+$ is that it only considers the *absolute* interference between pairs of samples. While it reflects the correlation of the $Q$-value changes between pairs of state-actions, it does not help us understand whether it is a *good* or *bad* correlation. Indeed, when function approximation is used, it is impossible for the learner to learn a good policy if generalized updates do not change the $Q$-values of other state-actions.

### 3.1.2 MIXED-GRADIENT INTERFERENCE FOR TEMPORAL DIFFERENCE LEARNING

To better distinguish good generalization and bad interference, we propose to quantify "how much the TD error decreases on another state-action if trained on one state-action." Let $\ell_\theta(s, a, s') =$

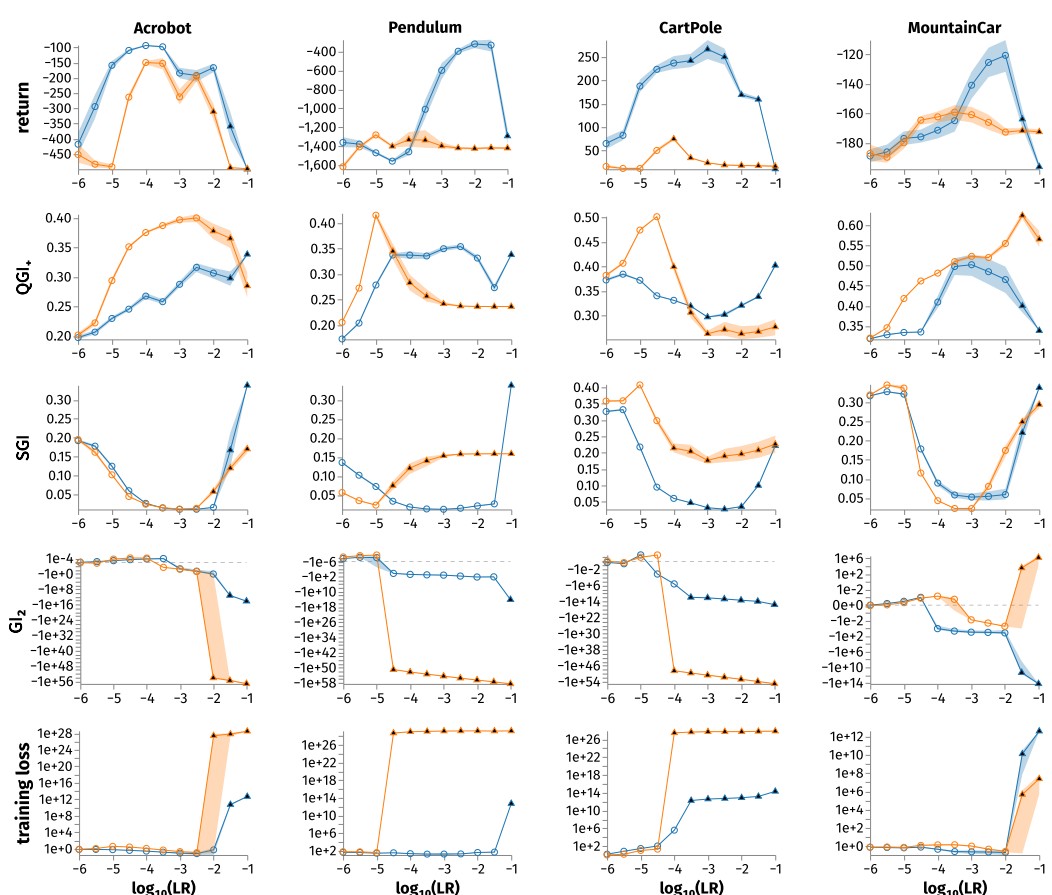

Figure 2: Every data point averages its metric (return, QGI$_+$, SGI, GI$_2$, or training loss) over 30 seeds, each seed averaging over all gradient steps of a training run. Comparing LN ⊙ vs. no normalization ⊙, QGI$_+$ and (more so) SGI positively correlate with the loss. GI$_2$ negatively correlates with the loss. The QGI$_+$ correlations tend to break once SGD has diverged so far that its weights are clamped (▲). Yet, for SGI and GI$_2$, their correlations tend to hold at any given learning rate (LR). MountainCar is an outlier for the correlation of GI$_2$. Shaded areas are 95% bootstrap CIs. These results use SGD. Adam results in Appendix F.

$(Q_\theta(s,a) - r(s,a) - \gamma \max_{a'} Q_\theta(s',a'))^2$. After training on $(s_t, a_t, s_t')$, the decrease of the TD error on $(s, a, s')$ is

$$\ell_{\theta_{t+1}}(s,a,s') - \ell_{\theta_t}(s,a,s') \approx \langle \theta_{t+1} - \theta_t, \nabla \ell_{\theta_t}(s,a,s') \rangle \qquad \text{(first-order approximation)}$$

$$= -\eta \langle \overset{\triangledown}{\nabla} \ell_{\theta_t}(s_t, a_t, s_t'), \nabla \ell_{\theta_t}(s,a,s') \rangle, \tag{7}$$

where the last equality is by Eq. (1). The last expression can be turned into an interference metric that uses an inner product between a semi-gradient and a full-gradient. This is different from the interference metric used in prior work such as Lyle et al. (2023), which uses an inner product between two semi-gradients, $\langle \overset{\triangledown}{\nabla} \ell_{\theta_t}(s_t, a_t, s_t'), \overset{\triangledown}{\nabla} \ell_{\theta_t}(s,a,s') \rangle$. We argue that the mixed "semi–full" version is more principled from the perspective of minimizing the loss (*cf.* Fujimoto et al. (2022)), as the "semi" part reflects how the parameters are updated, and the "full" part, loosely put, reflects the loss we want to minimize. We call the semi-gradients-only metric *semi-gradient interference* (SGI), and the mixed-gradient metric *mixed-gradient interference* (MGI).

Unlike QGI$_+$ in Section 3.1.1, SGI and MGI do not exclude the self-interference terms $(s,a) \neq (s_t, a_t)$. We include the self-interference terms because, following the loss minimization view, the self-interference terms are part of the first-order approximation. Again following the loss

minimization view, SGI and MGI do not take the absolute value, i.e. they use the cosine similarity $\cos(x, y) \triangleq \frac{\langle x, y \rangle}{\|x\|\|y\|}$ rather than $\cos_+$:

$$\mathsf{SGI} = \mathbb{E}_{(s,a,s'),(s_t,a_t,s'_t)\sim\mu}\left[\cos\left(\slashed{\nabla}\ell_{\theta_t}(s_t, a_t, s'_t),\ \slashed{\nabla}\ell_{\theta_t}(s, a, s')\right)\right], \tag{8}$$

$$\mathsf{MGI} = \mathbb{E}_{(s,a,s'),(s_t,a_t,s'_t)\sim\mu}\left[\cos\left(\slashed{\nabla}\ell_{\theta_t}(s_t, a_t, s'_t),\ \nabla\ell_{\theta_t}(s, a, s')\right)\right]. \tag{9}$$

Lyle et al. (2023) mentions a standard interpretation of elements of SGI (before taking the expectation): negative components indicate that "the network cannot reduce its loss on one subset without increasing its loss on another," a sign of low plasticity or low trainability that is generally bad for performance. Conversely, positive elements can indicate better generalization, as reducing the loss on a state-action helps reducing that of another. However, note that tabular $Q$-learning's $\mathsf{QGI}_+$ is zero, and its SGI and MGI are zero aside from the self-interference terms. Recall that this allows it to converge under milder conditions than algorithms like DQN.

**SGI and MGI also *positively* correlate with training loss in our experiments.** We show SGI in row three of Fig. 2, and MGI (whose results are qualitatively similar) in Appendix F. This is somewhat surprising: on one hand, this contrasts with the standard interference argument mentioned in Lyle et al. (2023), where positive gradient interference indicates generalization, a beneficial property. On the other hand, this aligns with the empirical findings of Lyle et al. (2023; 2025) that gradient interference magnitudes (i.e. the absolute values) can negatively correlate with task performance for non-stationary settings.

This also aligns with the combined observations that LN is known to (i) empirically improve returns, and (ii) theoretically and empirically increase isometry (Appendix C). Higher isometry suggests to some extent that training is closer to tabular RL, with lessened interference and generalization.

An additional, speculative explanation for the correlation between gradient interference and loss is overshooting, where training updates move in the correct direction, but too far. This overshooting may cause instability (Mahmood, 2010; Dabney & Barto, 2012; Mahmood et al., 2012; McLeod et al., 2021; Kearney, 2023; Javed et al., 2025). In any case, this finding that first-order in-distribution generalization *positively* correlates with the loss remains somewhat counterintuitive. In the following sections, we ultimately aim to more precisely explain this finding, and to come up with a more intuitive metric.

## 3.2 A Refined Analysis

In this subsection, we provide more supporting theory for the empirical findings in Section 3.1.

The empirical findings in Section 3.1 suggests that both the *absolute* interference metrics ($\mathsf{QGI}_+$) and the *signed* interference metrics (SGI and MGI) correlate positively with the training loss. While the former aligns with our hypothesis, the latter does not. In fact, a similar mismatch between the intuition and the empirical finding also appears in Lyle et al. (2023).

We argue that the first-order approximation on the loss decrement we employ in Section 3.1.2 has to be refined. If we perform similar approximation as Eq. (7) on the loss decrement, but up to the *second order*, then we get

$$\ell_{\theta_{t+1}}(s, a, s') - \ell_{\theta_t}(s, a, s')$$

$$\approx \langle \theta_{t+1} - \theta_t, \nabla\ell_{\theta_t}(s, a, s') \rangle + \tfrac{1}{2}(\theta_{t+1} - \theta_t)^\top \left[\nabla^2\ell_{\theta_t}(s, a, s')\right](\theta_{t+1} - \theta_t) \tag{10}$$

$$= \underbrace{-\eta\langle \slashed{\nabla}\ell_{\theta_t}(s_t, a_t, s'_t), \nabla\ell_{\theta_t}(s, a, s') \rangle}_{\text{first-order term}} + \underbrace{\tfrac{1}{2}\eta^2 \slashed{\nabla}\ell_{\theta_t}(s_t, a_t, s'_t)^\top \left[\nabla^2\ell_{\theta_t}(s, a, s')\right] \slashed{\nabla}\ell_{\theta_t}(s_t, a_t, s'_t)}_{\text{second-order term } (\star)}.$$

Recall $\ell_\theta(s, a, s') = (Q_\theta(s, a) - r(s, a) - \gamma\max_{a'} Q_\theta(s', a'))^2$, and $\Delta_\theta(s, a, s') = Q_\theta(s, a) - r(s, a) - \gamma\max_{a'} Q_\theta(s', a')$. Direct calculation on the Hessian of the loss function gives:

$$\tfrac{1}{2}\nabla^2\ell_{\theta_t}(s, a, s') = \frac{1}{\ell_{\theta_t}(s, a, s')}\nabla\ell_{\theta_t}(s, a, s')\nabla\ell_{\theta_t}(s, a, s')^\top + \Delta_{\theta_t}(s, a, s')\nabla^2 h_{\theta_t}(s, a, s'),$$

where we denote $h_\theta(s, a, s') = Q_\theta(s, a) - \gamma\max_{a'} Q_\theta(s', a')$. Using this in the second-order term $(\star)$ above, we get

$$(\star) = \eta^2 \frac{\langle \slashed{\nabla}\ell_{\theta_t}(s_t, a_t, s'_t), \nabla\ell_{\theta_t}(s, a, s') \rangle^2}{\ell_{\theta_t}(s, a, s')}$$

$$+ \eta^2 \Delta_{\theta_t}(s,a,s') \not\nabla \ell_{\theta_t}(s_t,a_t,s'_t)^\top \nabla^2 h_{\theta_t}(s,a,s') \not\nabla \ell_{\theta_t}(s_t,a_t,s'_t).$$

While the second part above is related to the second-order derivative of $Q_\theta$, the first part is always non-negative and is exactly the *squared interference* (from Eq. (7)) scaled by the loss inverse.

Therefore, we argue that *signed* interference and *absolute/squared* interference have to be combined, along with the second-order shape of the $Q$-network, to give a more complete picture for the stability of deep $Q$-learning. This more complete analysis also suggests that the counterintuitive empirical results for SGI and MGI might be because they only reflect the first-order term in the loss decrement, but neglect the effect from the second-order term. The second-order term might dominate when the learning rate is too large compared to the smoothness of the loss, causing the overshooting we previously discuss. This may align with claims that LN smooths the loss landscape (Lee et al., 2023).

Measuring this second-order approximation rather than the first-order approximation, MGI, is a natural next step. For simplicity, we defer to future work the study of the more complicated, second term in ($\star$). We combine the first-order term interference, MGI, with only the first term from ($\star$). We refer to this second-order approximation of the gradient interference as $\mathsf{GI}_2$:

$$\mathsf{GI}_2 = 2\eta \, \mathbb{E}_{(s,a,s'),(s_t,a_t,s'_t)\sim\mu} \big[ \langle \not\nabla \ell_{\theta_t}(s_t,a_t,s'_t), \, \nabla \ell_{\theta_t}(s,a,s') \rangle \big]$$
$$- \eta^2 \mathbb{E}_{(s,a,s'),(s_t,a_t,s'_t)\sim\mu} \left[ \frac{1}{\ell_{\theta_t}(s,a,s')} \, \langle \not\nabla \ell_{\theta_t}(s_t,a_t,s'_t), \, \nabla \ell_{\theta_t}(s,a,s') \rangle^2 \right]. \quad (11)$$

$\mathsf{GI}_2$ could be further specified as $\mathsf{MGI}_2$, but, given our loss minimization argument, we do not measure the equivalent with semi-gradients only (which could be specified as $\mathsf{SGI}_2$).

**$\mathsf{GI}_2$ *negatively* correlates with training loss in our experiments (fourth row of Fig. 2).** This aligns with our hypothesis, though leaves unanswered the question of which difference from MGI is most important for negating the empirical correlation with the loss: (i) the second-order term; or (ii) the use of the dot product alone, rather than cosine similarity. That is, compared to MGI, the metric $\mathsf{GI}_2$ not only (i) includes a $\langle \not\nabla \ell_{\theta_t}(s_t,a_t,s'_t), \nabla \ell_{\theta_t}(s,a,s') \rangle^2$ term; but also (ii) lacks the $(\|\not\nabla \ell_{\theta_t}(s_t,a_t,s'_t)\| \|\nabla \ell_{\theta_t}(s,a,s')\|)^{-1}$ factor in its first-order term. However, we find empirically that the first-order dot-product term alone, which we call DGI (dot-product gradient interference), still correlates positively with the loss (which we show in Appendix F). This suggests the second-order term is indeed more important than the first-order term in the setting we study.

### 3.3 How Robustly Does LN Improve $\mathsf{GI}_2$?

We would like to find the most basic explanation for why LN improves deep $Q$-learning. One avenue for better establishing that $\mathsf{GI}_2$ may be an important explanation, rather than a side effect, would be to show that LN improves $\mathsf{GI}_2$ even when temporarily added to a training run without LN. That is, to only add LN periodically throughout training, just to see if it immediately improves $\mathsf{GI}_2$ at that training step, without any actual training. Remarkably, we indeed find that even LN improves $\mathsf{GI}_2$ even without any retraining (Fig. 3).

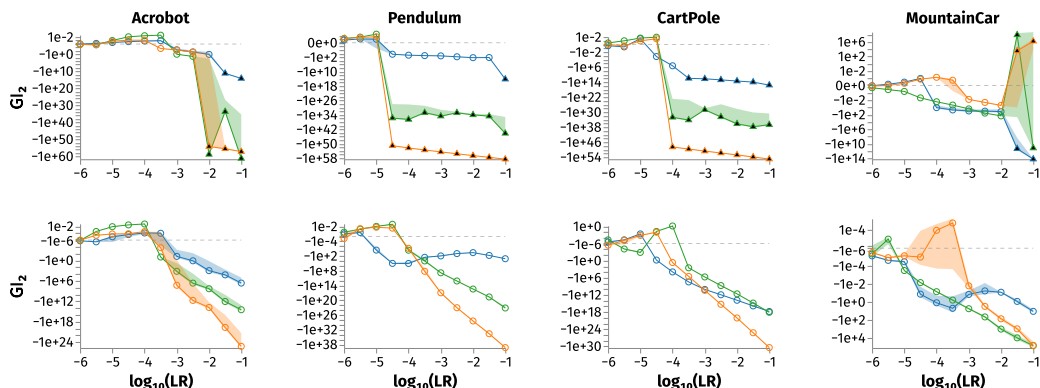

Figure 3: We use SGD (top row) and Adam (bottom row) to train without normalization again, but this time we add LN temporarily ⭕, periodically throughout training, only to measure $\mathsf{GI}_2$. For comparison, we again show LN ⭕ and no normalization ⭕ as well. For both SGD and Adam, LN *still* tends to improve $\mathsf{GI}_2$, even without allowing the network to take even a single training step with LN in place. MountainCar is again an exception.

### 3.4 PROVABLE BENEFITS OF NORMALIZATION

As the empirical and theoretical analyses in Section 3.1, Section 3.2, and Section 3.3 suggest, the squared interference dominates the stability of deep $Q$-learning, and it is related to the second-order approximation of the loss decrement. However, we have not understood *why* normalization stabilizes or accelerates training. In fact, even for the simpler case of supervised learning, we are not aware of a compelling theoretical explanation in the literature, though we suspect that our analysis is not likely to be novel.

In this subsection we provide some theoretical evidence on the effect of feature normalization in stochastic gradient descent (SGD) for *linear regression*, a special case of linear $Q$-learning, the simplest form of $Q$-learning with function approximation. The problem setting is formally defined in Definition 1. In particular, we show that feature normalization improves an SGD progress–stability tradeoff. This is offered only as supporting context for our second-order interference view, not as a claim of novelty.

**Definition 1** (Stochastic gradient descent for linear regression). *Let $(x, y) \in \mathbb{R}^d \times \mathbb{R}$ be drawn from a fixed distribution $\mu$ and let the goal be to minimize $L(\theta) = \mathbb{E}_{(x,y)\sim\mu}[(x^\top \theta - y)^2]$. With SGD, at each iteration $t$, a sample $(x_t, y_t) \sim \mu$ is drawn and the parameter is updated as $\theta_{t+1} = \theta_t - \eta x_t(x_t^\top \theta_t - y_t)$.*

For simplicity, we assume *realizability*:

**Assumption 1** (realizability). *There exists a $\theta^\star \in \mathbb{R}^d$ such that $y = x^\top \theta^\star$.*

Although Assumption 1 assumes noiseless feedback, this is only to simplify our exposition. It is straightforward to extend it to the case with noise scale $\mathbb{E}[(y - x^\top \theta^\star)|x] = \sigma^2(x)$ for some function $\sigma(x)$ increasing in $\|x\|$.

We consider two quantities that are usually used to measure the progress of training. One is $R(\theta) = \|\theta - \theta^\star\|^2$, the squared distance to the optimal solution. The other is $L(\theta)$, the expected loss defined in Definition 1.

Under Assumption 1, the update of SGD with learning rate $\eta$ yields a expected distance decrement equality:

$$\mathbb{E}[R(\theta_{t+1})|\theta_t] = R(\theta_t) - 2\eta A_t + \eta^2 B_t,$$

for some $A_t, B_t > 0$ that depends on data distribution $\mu$ and $\theta_t$. They are the first-order and second-order terms in optimization, similar to Eq. (10). We can prove the following:

**Theorem 2.** *Under fixed $R(\theta_t)$, $A_t^2/B_t$ is maximized when $\|x\|$ is a constant in the data distribution.*

There are two different ways to interpret Theorem 2:

- With the optimal choice of the learning rate $\eta$, the distance decrement $R(\theta_t) - \mathbb{E}[R(\theta_{t+1})|\theta_t]$ is maximized when $\|x\|$ are all equal. This is because $\max_\eta\{2\eta A_t - \eta^2 B_t\} = A_t^2/B_t^2$.

- Under a fixed first-order term $\eta A_t$, the second-order term $\eta^2 B_t$ is minimized when $\|x\|$ are all equal. This is because the second-order term $\eta^2 B_t$ is equal to $F^2 B_t/A_t^2$ where $F = \eta A_t$ is the first-order term. Similarly, under a fixed second-order term, the first-order term is maximized when $\|x\|$ are all equal.

The first interpretation indicates that feature normalization (i.e., making $\|x\|$ all equal) can achieve the most decrement in distance to $\theta^\star$, while the second interpretation indicates that feature normalization always achieves the best possible trade-off between the two terms, under any learning rate.

For the other measurement $L(\theta)$, we can also prove a similar property, under an additional assumption that $\|x\|$ (scale) and $\frac{x}{\|x\|}$ (direction) are independent under the data distribution. Similarly, the following loss decrement equality holds for SGD:

$$\mathbb{E}[L(\theta_{t+1})|\theta_t] = L(\theta_t) - 2\eta C_t + \eta^2 D_t,$$

for some $C_t, D_t > 0$ that depends on $\mu$ and $\theta_t$, and we have

**Theorem 3.** *Assume that $\|x\|$ and $\frac{x}{\|x\|}$ are independent under $\mu$. Then under fixed $L(\theta_t)$, $C_t^2/D_t$ is maximized when $\|x\|$ is a constant in the data distribution.*

We have the similar two ways to interpret Theorem 3 as in Theorem 2.

Extending Theorem 2 and Theorem 3 to temporal-difference learning requires several additional assumptions, which would make the analysis diverge further from practice. We therefore leave it as future work to seek a better perspective for understanding normalization in TD from the viewpoint of optimization theory. While linear regression is not the same as TD, it more closely resembles deep $Q$-learning with a target network, where the regression target changes slowly or remains fixed except for periodic updates. Our theory suggests that with a target network, normalization accelerates convergence in the regression problem defined by the target. Consequently, it could tolerate faster target network updates than the version without a target network. In the extreme, updating the target network at every step reduces to temporal-difference learning.

## 4 LIMITATIONS AND FUTURE DIRECTIONS

One path for future theoretical work is to extend the linear regression theory to linear temporal difference learning, with the goal of proving the connection between gradient interference metrics and convergence speed. Another path is to connect our theory with Gallici et al. (2025). Maybe the most important empirical directions are to experiment with a wider variety of environments, including image-based observations and continuous action spaces. Finally, both theoretically and empirically, an important direction is to consider the effects of model scaling.

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

## A    RELATED WORK

A broad family of normalization techniques has played an important role in stabilizing the training of neural network, including LN (Ba et al., 2016), batch normalization (Ioffe & Szegedy, 2015), weight normalization (Salimans & Kingma, 2016), spectral normalization (Miyato et al., 2018), and group normalization (Wu & He, 2018). The significant practical success of these methods has motivated a growing body of theoretical work aimed at understanding their mechanisms, providing various insights on the mechanism of different normalization modules. In this work, we are specifically interested in understanding whether and how these mechanisms carry to the RL setting.

**Adaptive learning rate.**    Arora et al. (2018) argue that normalization effectively adapts the learning rate during training. This property is common across normalization methods, as they impose invariance of training loss to weight scalings. Such invariance induces a monotonic growth in the weight norms during training and has been central to the development of the weight normalization method (Salimans & Kingma, 2016). Lyle et al. (2024) study the similar phenomenon in reinforcement learning, but focus on its side effect in the multi-task setting caused by the vanishing learning rate. Our experiments, focusing on the single-task setting, show that the automatic adjustment of the learning rate is an important property in DQN which could improve performance. However, this alone cannot fully explain why other normalization layers, such as weight normalization, are less effective in DQN.

**Faster regression.**    Kohler et al. (2019) prove batch normalization can accelerate linear regression when the input data is Gaussian. In this simplified setting, they provide quantitative convergence bounds for linear regression with batch normalization, demonstrating that gradient descent converges more rapidly with normalization. The acceleration arises from decoupling the optimization of weight magnitudes from their directions. However, their analysis relies heavily on the Gaussian assumption. In our work, we extend this proof to hold under substantially weaker assumptions on the data distribution.

**Isometry.** Joudaki et al. (2023) introduce the notion of isometry, which quantifies the degree of orthogonality among samples or features. Their study shows that normalization layers induce an implicit bias toward orthogonalizing data representations in deep random neural networks. This contrasts with networks without normalization, where data samples become increasingly aligned as network depth grows (Daneshmand et al., 2021). While deep neural networks are widely used in supervised learning, DQN primarily relies on relatively shallow networks. Therefore, it remains unclear whether isometry can fully explain the mechanisms underlying DQN.

**Gradient explosion with batch normalization.** Despite its advantages, batch normalization can also introduce significant drawbacks that may limit its applicability. Yang et al. (2019) prove that batch normalization leads to gradient explosion as network depth increases, a phenomenon initially thought to be unavoidable. However, Meterez et al. (2024) demonstrate that gradient explosion can in fact be mitigated through careful choices of hyperparameters and initialization.

**Normalization in reinforcement learning.** An important open question is whether explanations for supervised learning extend to reinforcement learning settings. Addressing this is crucial, as some normalization modules do not transfer effectively across different learning paradigms. For example, batch normalization not only enhances the training of transformers but can also destabilize it (Shen et al., 2020). Interestingly, a similar pattern is observed in DQN, where only LN has consistently been shown to be effective. For example, Hiraoka et al. (2022); Lee et al. (2023); Ball et al. (2023); Elsayed et al. (2024); Gallici et al. (2025); Lee et al. (2025) all proposed combinations of techniques to enhance deep RL algorithms, and all include LN as an important component. Among them, Ball et al. (2023) argue that LN improves the landscape of $Q$-function, preventing catastrophic value extrapolation, and Lee et al. (2023) argues that LN makes the loss landscape smoother, avoiding the loss of plasticity.

Besides LN, attempts to use other normalization schemes have been made. Bhatt et al. (2024) propose an adaptation of batch normalization to DQN (called CrossQ) that addresses the issue of distribution shift of naive application of BatchNorm to DQN. Palenicek et al. (2025) further enhance CrossQ by combining it with weight normalization. On the other hand, Gogianu et al. (2021) argue that spectral normalization provides automatic learning rate adaptation that benefits the training dynamics. Overall, however, other normalization schemes in DQN are rarer than LN.

## B POSTPONED PROOFS

**Standing assumptions for this section.** Throughout this section, expectations are conditional on $\theta_t$ (equivalently on $e_t = \theta_t - \theta^\star$) and taken over fresh i.i.d. samples. We assume: (i) finite moments and nondegeneracy, $\mathbb{E}\|x\|^4 < \infty$ and $\mathbb{E}\|x\|^2 > 0$; (ii) the magnitude–direction model $x = az$ with $a = \|x\| \geq 0$ and $z = x/\|x\|$ when $a > 0$, with $a$ independent of $z$ (and $z$ defined arbitrarily on $\{a = 0\}$); (iii) when $(x, \hat{x})$ appear together they are independent i.i.d. draws, each factorizable as $az$ with the same independence structure.

*Proof of Theorem* **??**. Under Assumption 1, let $e_t \triangleq \theta_t - \theta^\star$. One SGD step gives $e_{t+1} = e_t - \eta\, x_t x_t^\top e_t$, hence

$$\mathbb{E}\big[\|e_{t+1}\|^2 \mid \theta_t\big] = \|e_t\|^2 - 2\eta A_t + \eta^2 B_t,$$

with

$$A_t \triangleq e_t^\top \mathbb{E}[xx^\top]e_t, \qquad B_t \triangleq e_t^\top \mathbb{E}[xx^\top xx^\top]e_t.$$

Write $x = az$ with $a = \|x\|$ and $z = x/\|x\|$. Then

$$A_t = \mathbb{E}[a^2 v^2], \qquad B_t = \mathbb{E}[a^4 v^2], \quad \text{where } v \triangleq z^\top e_t.$$

By Cauchy–Schwarz applied to the measure with density $v^2$ (i.e., with $Y = a^2$ and $W = v^2$ so that $(\mathbb{E}[YW])^2/\mathbb{E}[Y^2W] \leq \mathbb{E}[W]$),

$$\frac{A_t^2}{B_t} = \frac{\big(\mathbb{E}[a^2 v^2]\big)^2}{\mathbb{E}[a^4 v^2]} \leq \mathbb{E}[v^2] = e_t^\top \mathbb{E}[zz^\top]e_t,$$

with equality if and only if $a^2$ is almost surely constant on $\{v \neq 0\}$ (it may vary on $\{v = 0\}$ without effect). Therefore, for fixed $R(\theta_t) = \|e_t\|^2$, the ratio $A_t^2/B_t$ is maximized when $\|x\|$ is constant under $\mu$. $\square$

*Proof of Theorem 3.* Let $\mathbb{E}_{(x,y)}$ denote $\mathbb{E}_{(x,y)\sim\mu}$ and define $\varepsilon_t(x,y) \triangleq x^\top \theta_t - y$. Expanding one SGD step yields

$$L(\theta_{t+1}) = \mathbb{E}_{(x,y)}\Big[\big(\varepsilon_t(x,y) - \eta\, x^\top x_t\, \varepsilon_t(x_t, y_t)\big)^2\Big]$$

$$= L(\theta_t) - 2\eta\, \mathbb{E}_{(x,y)}\big[\varepsilon_t(x,y)\, \varepsilon_t(x_t, y_t)\, x^\top x_t\big] + \eta^2\, \mathbb{E}_{(x,y)}\big[\varepsilon_t(x_t, y_t)^2\, (x^\top x_t)^2\big].$$

Taking expectation over $(x_t, y_t) \sim \mu$, we obtain

$$\mathbb{E}[L(\theta_{t+1}) \mid \theta_t] = L(\theta_t) - 2\eta\, C_t + \eta^2 D_t,$$

where

$$C_t \triangleq \mathbb{E}_{(x,y)}\mathbb{E}_{(\hat{x},\hat{y})}\big[\varepsilon_t(x,y)\, \varepsilon_t(\hat{x},\hat{y})\, x^\top \hat{x}\big],$$

$$D_t \triangleq \mathbb{E}_{(x,y)}\mathbb{E}_{(\hat{x},\hat{y})}\big[\varepsilon_t(\hat{x},\hat{y})^2\, (x^\top \hat{x})^2\big].$$

Thus, for any $\eta$, the loss decrement has the quadratic form above, and the ratio governing the optimal decrement is $C_t^2/D_t$.

Under Assumption 1, write $\varepsilon_t(x,y) = x^\top e_t$ with $e_t \triangleq \theta_t - \theta^\star$. Decompose $x = az$ and $\hat{x} = \hat{a}\,\hat{z}$ with $a = \|x\|$, $z = x/\|x\|$. Using independence between magnitude and direction and i.i.d. sampling across the two draws, define

$$U_t := \mathbb{E}_{z,\hat{z}}\big[(z^\top e_t)(\hat{z}^\top e_t)(z^\top \hat{z})\big],$$

$$W_t := \mathbb{E}_{z,\hat{z}}\big[(\hat{z}^\top e_t)^2(z^\top \hat{z})^2\big].$$

Then

$$C_t = \mathbb{E}\big[(a\, z^\top e_t)(\hat{a}\,\hat{z}^\top e_t)(a\hat{a}\, z^\top \hat{z})\big] = \mathbb{E}[a^2]^2\, U_t,$$

$$D_t = \mathbb{E}\big[(\hat{a}\,\hat{z}^\top e_t)^2(a\hat{a}\, z^\top \hat{z})^2\big] = \mathbb{E}[a^2]\,\mathbb{E}[a^4]\, W_t.$$

Consequently,

$$\frac{C_t^2}{D_t} = \frac{\mathbb{E}[a^2]^4}{\mathbb{E}[a^2]\,\mathbb{E}[a^4]} \cdot \frac{U_t^2}{W_t}.$$

For fixed $L(\theta_t) = \mathbb{E}[\varepsilon_t(x,y)^2] = \mathbb{E}[a^2]\,\mathbb{E}[(z^\top e_t)^2]$, the directional term $\mathbb{E}[(z^\top e_t)^2]$ is fixed, hence $\mathbb{E}[a^2]$ is pinned. Thus maximizing $C_t^2/D_t$ over the magnitude distribution reduces to minimizing $\mathbb{E}[a^4]$, which is achieved when $a^2$ is almost surely constant (by Jensen). Hence $C_t^2/D_t$ is maximized when $\|x\|$ is constant under $\mu$. $\square$

**Degenerate edge cases.** If $e_t = 0$ or $z^\top e_t = 0$ almost surely, then $A_t = C_t = 0$ and the one-step maximization statements are vacuous. If $\mathbb{P}(a = 0) > 0$, values of $z$ on $\{a = 0\}$ are irrelevant since all weighted terms place zero mass there.

## C  ISOMETRY

**Theoretical hypothesis.** Let $X = \{x_1, x_2 \ldots, x_n\} \subset \mathbb{R}^d$. The Gram matrix $G(X)$ is defined as $G(X)_{ij} = \langle x_i, x_j \rangle, \forall i, j \in [n]$. Following Joudaki et al. (2023), the *isometry* of $X$ is defined as

$$\mathsf{ISO}\,(X) = \det(G(X))^{\frac{1}{|X|}} \Big/ \big(\tfrac{1}{|X|}\,\mathrm{Tr}(G(X))\big),$$

where $|X| = n$ is the cardinality of $X$. $\mathsf{ISO}(X)$ is at most 1 and captures the degree of isotropy: $\det(G(X))$ is equal to the squared volume of the parallelotope spanned by $x_1, \ldots, x_n$, and $\mathrm{Tr}(G(X)) = \sum_{i=1}^{n} \|x_i\|^2$ is sum of their squared norms. The closer this ratio is to 1, the more isotropic $X$ is.

To extend this notion to distributions, we define the order-$k$ isometry of a distribution $\mu$ as

$$\mathsf{ISO}_k\left(\mu\right) = \mathbb{E}_{X=\{x_1,\dots,x_k\}\sim\mu^k}\left[\mathsf{ISO}(X)\right],$$

where $X$ consists of $k$ independent samples from $\mu$. This quantity measures the expected isometry of size-$k$ datasets. Note that $\det(G(X)) = 0$ whenever the vectors in $X$ are linearly dependent, so $k$ should not be chosen larger the dimension of the subspace spanned by the support of $\mu$.

Let $z \sim \overline{\mu}$ be the distribution generated by first drawing $x \sim \mu$ and setting $z = x/\|x\|$. It is straightforward to show that for any $k$, $\mathsf{ISO}_k(\mu) \leq \mathsf{ISO}_k(\overline{\mu})$, given Joudaki et al. (2023)'s theorem:

**Theorem 4** (Theorem 1 of Joudaki et al. (2023)). *Let* $X = \{x_1, \dots, x_n\}$ *and be an arbitrary dataset and let* $Z = \{x_1/\|x_1\|, \dots, x_n/\|x_n\|\}$. *Then* $\mathsf{ISO}(X) \leq \mathsf{ISO}(Z)$.

Theorem 4 implies that for any size-$k$ datasets $X$ and its normalized version $Z$, $\mathsf{ISO}(X) \leq \mathsf{ISO}(Z)$. The inequality $\mathsf{ISO}_k(\mu) \leq \mathsf{ISO}_k(\overline{\mu})$ follows by taking expectation over the data distribution $\mu$.

Joudaki et al. (2023) provide empirical evidence that, in supervised learning, LN results in increased ISO of the network's intermediate features throughout training. They also empirically find that ISO positively correlates with the convergence rate. We hypothesize that LN will similarly increase $\mathsf{ISO}_k$, even after training, in deep $Q$-learning. We likewise hypothesize that $\mathsf{ISO}_k$ will correlate with returns.

**Empirical evidence.** Beyond the fact that $k$ should not be chosen larger than the rank of the Gram matrix, it is unclear what $k$ should be set to. There are several potential options: setting $k$ as the dimensionality of the observation space of the environment, or the total dimensionality of the inputs (either the observation dimensionality plus the action dimensionality, or the observation dimensionality times the action dimensionality, or a small, fixed constant such as 3. We find that all such $k$ values (not shown) give results aligning fairly well with our hypotheses. We show $\mathsf{ISO}_\mathsf{p}$, which sets $k$ to the observation **p**lus action dimensionality, in Fig. 4. We show results for Adam and for all other $k$ values in Appendix E.

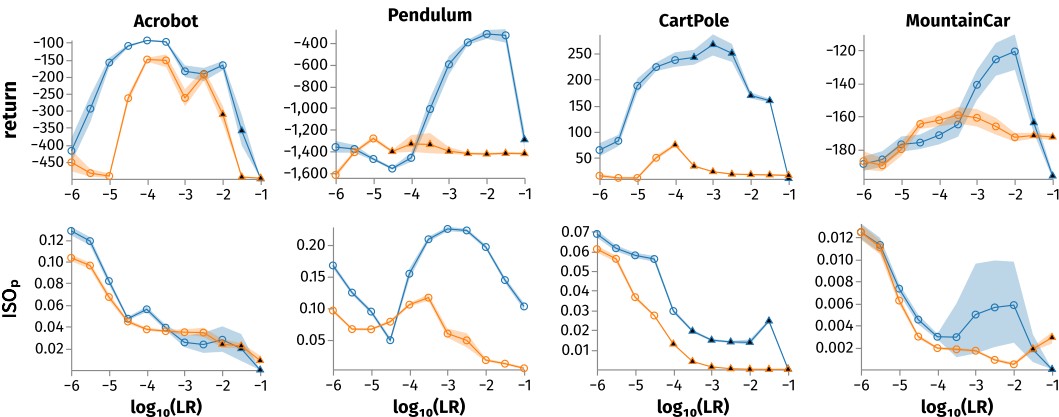

Figure 4: $\mathsf{ISO}_p$ correlates well with return. These correlations tend to hold at any given learning rate (LR), and also sometimes across learning rates. SGD results only. Results with Adam and other values of $k$ (for $\mathsf{ISO}_k$) are in Appendix E.

**Future work.** Theoretically connecting the isometry metric to the convergence rate (even in supervised learning) is perhaps the most important open question here.

## D  IMPLICIT LEARNING RATE DECAY

**Theoretical hypothesis.** In this subsection, we focus on the *effective learning rate* for the first layer of our neural network, i.e., the $W$ parameter in Eq. (3). Notice that the output value will not change if we replace the $W$ in Eq. (3) with $\alpha W$ for any $\alpha > 0$ as the function LN is invariant to the scale of $W$.

Now, we review general properties of this type of functions. Assume $f(\alpha w) = f(w)$ for any $\alpha > 0$. Since $f(w)$ is invariant to the scale of $w$, a meaningful update on $f(w)$ must make an update in the unit vector $\hat{w} = \frac{w}{\|w\|}$ that represents the *direction*. The following theorem by Arora et al. (2018) shows (i) that the effective learning rate in the space of $\hat{w}$ is roughly $\frac{\eta}{\|w\|^2}$, and (ii) that $\|w\|$ monotonically increases over time. Jointly, they imply that the effective learning rate decreases over time.

**Lemma 5** (Theorem 2.5 of Arora et al. (2018)). *Let $f(w)$ be invariant to the scale of $w$. Denote $\hat{w} \triangleq \frac{w}{\|w\|}$. For the gradient update $w^+ = w - \eta \nabla_w f(w)$, we have $\hat{w}^+ = \hat{w} - \frac{\eta}{\|w\|^2} \nabla_{\hat{w}} f(\hat{w}) + O(\eta^2)$ and $\|w^+\|^2 = \|w\|^2 + \frac{\eta^2}{\|w\|^2} \|\nabla_{\hat{w}} f(\hat{w})\|^2$.*

Such a learning rate decay argument not only applies to LN, but also to other forms of normalization such as batch normalization and weight normalization. Lyle et al. (2024) observe that such an implicit learning rate decay could be harmful to continual reinforcement learning, as the learner becomes slower and slower in learning new tasks. To address this, they periodically project $W$ back to a ball of fixed radius.

**Empirical evidence.** In preliminary experiments (not shown), following Lyle et al. (2024), we tested removing LN's implicit learning rate decay by projecting the weights to back to their initial norm after every gradient step. We also tested explicitly adding LN's implicit learning rate decay (i.e., setting the learning rate based on the norm of the weights) to a network without normalization. In both cases, aligning with (Lyle et al., 2024), our results suggested that implicit learning rate decay might sometimes increase returns, but did not fully account for LN's increase.

# E    ALL ISO$_k$ PLOTS

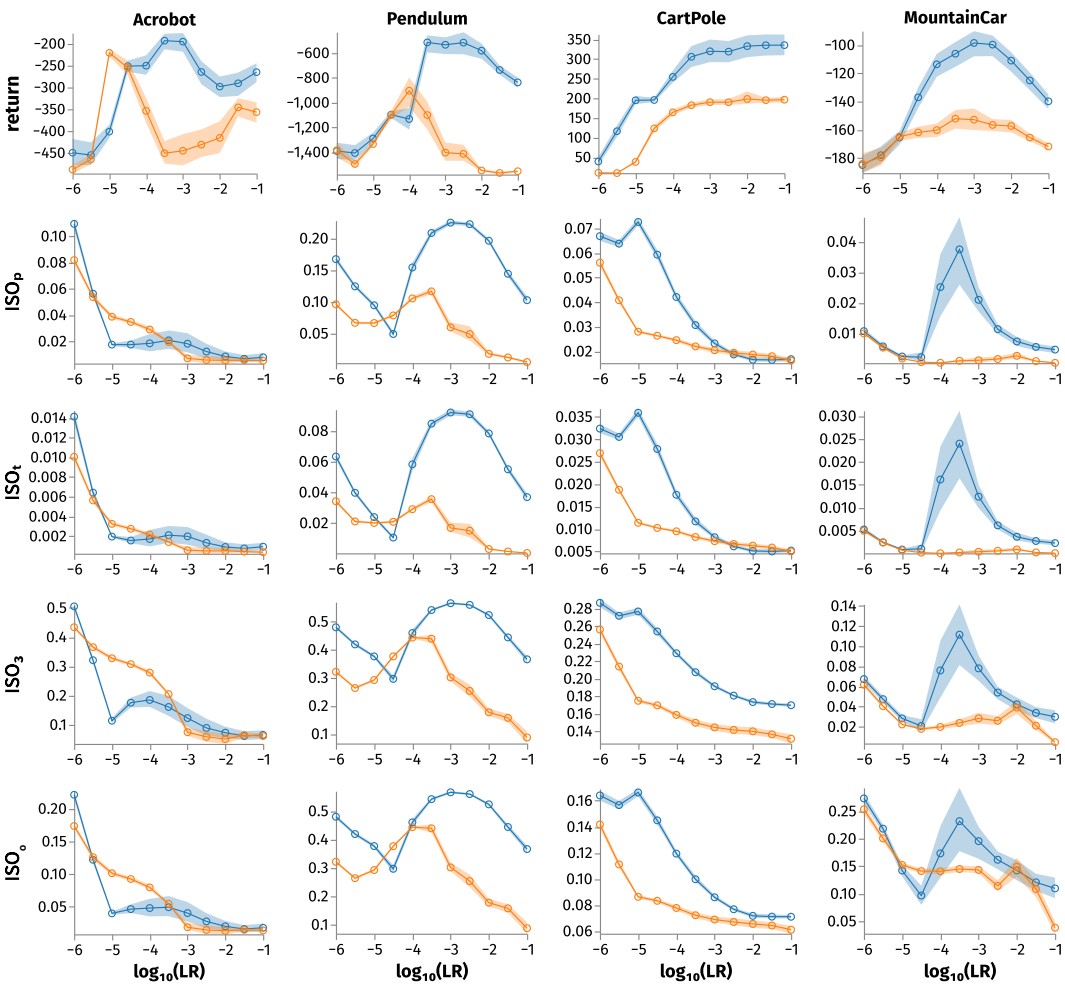

Figure 5: The same as Fig. 4, but using Adam instead of SGD, and including all of the $k$ values mentioned in the main text for ISO$_k$. The results are mostly similar at all values of $k$. ISO$_t$ sets $k$ to the dimensionality of the observation space **t**imes the dimensionality of the action space. ISO$_3$ sets $k$ to 3. ISO$_o$ sets $k$ to the dimensionality of the **o**bservation space alone.

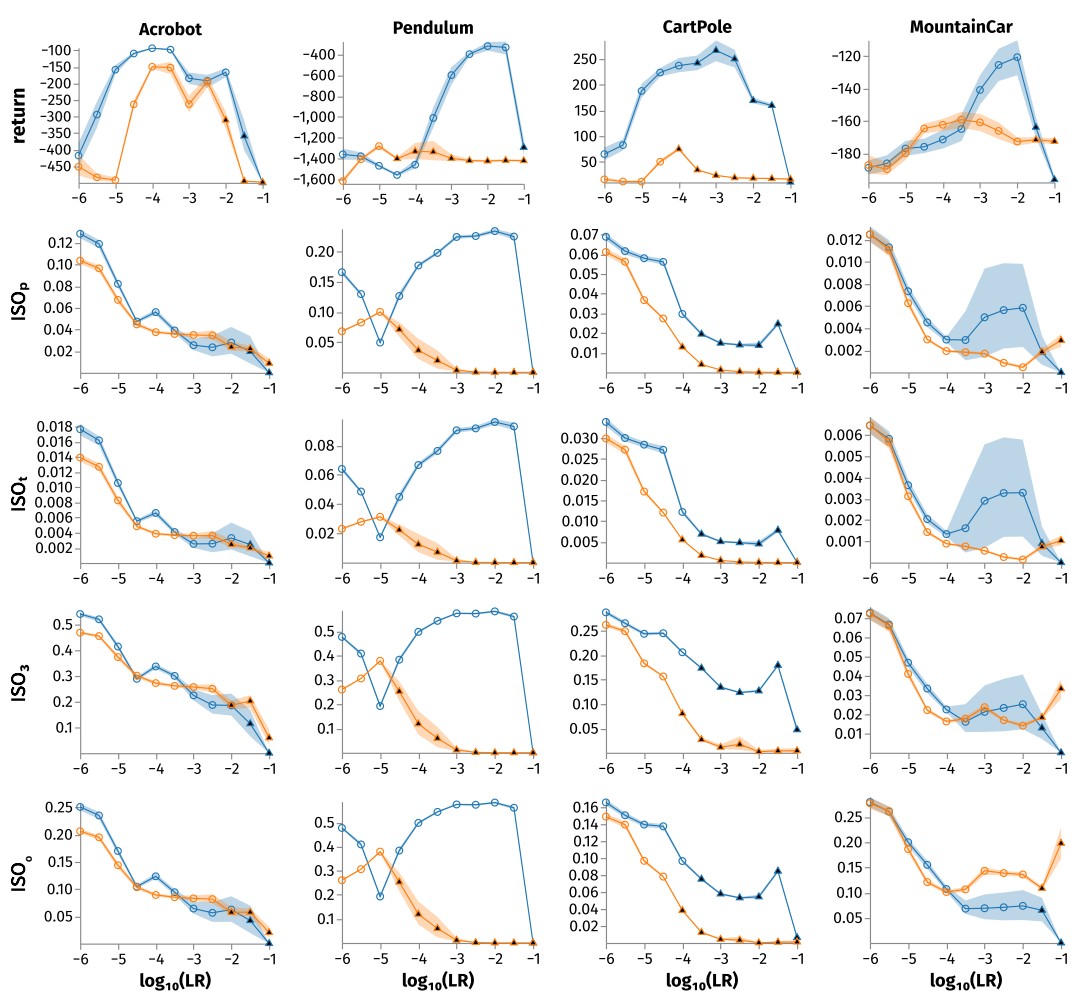

Figure 6: The same as Fig. 5, but using SGD. The various values of $k$ again give mostly similar results.

# F  ALL PER-LEARNING RATE GRADIENT INTERFERENCE PLOTS

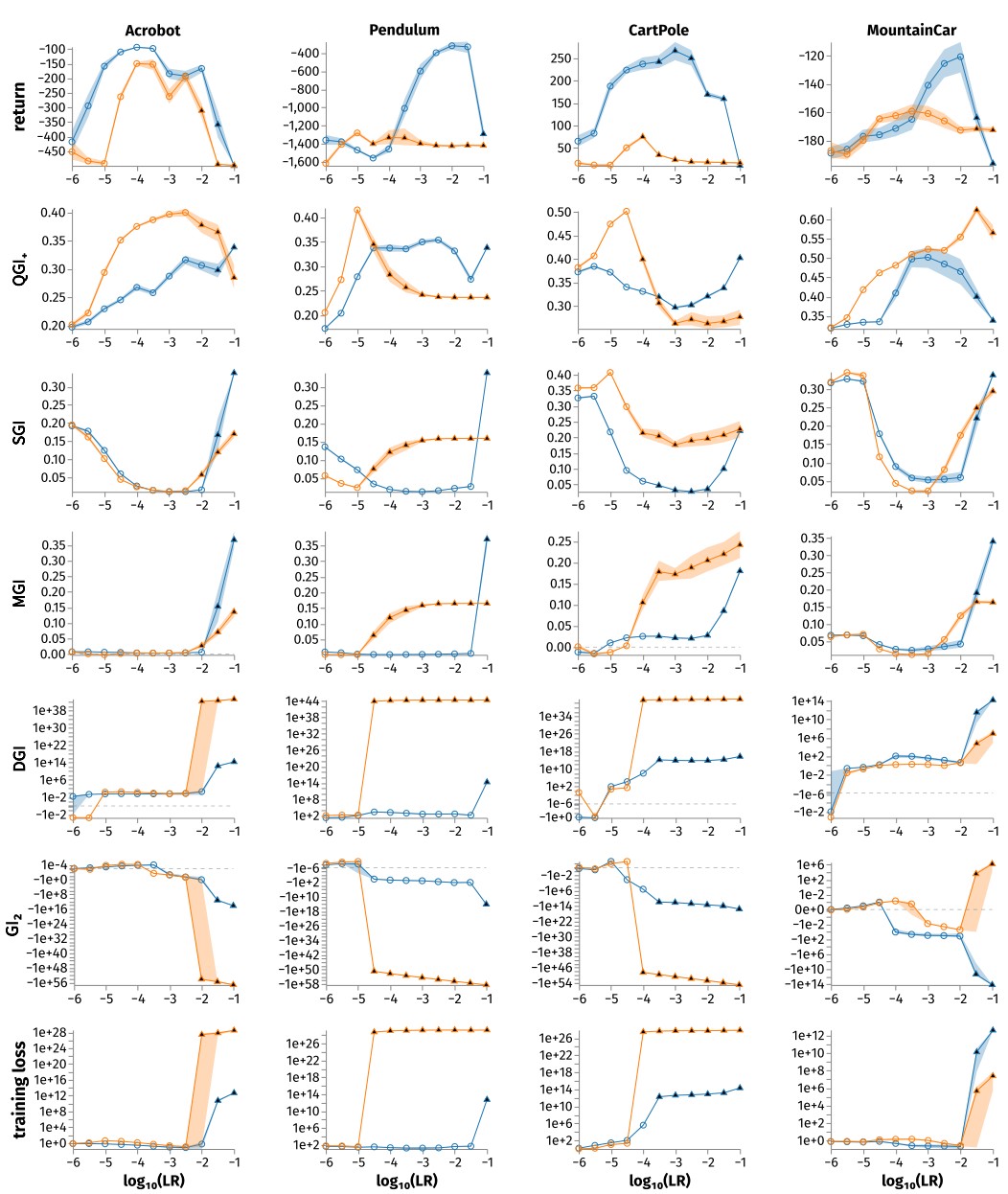

Figure 7: All of the gradient interference metrics when training with SGD. The return, training loss, QGI$_+$, and SGI are the same as plots as in Fig. 2. MGI is the mixed-gradient interference, i.e. the cosine similarity of each pair of semi-gradients and full-gradients of the TD error. As mentioned in Section 3.1.2, there are theoretical arguments for focusing on MGI instead of SGI, though in our experiments it still counterintuitively correlates with return. DGI is the dot-product gradient interference, the dot product version of MGI. That is, DGI is the dot product, rather than the cosine similarity, of each semi-gradient and full-gradient. As mentioned in the main text, DGI still positively correlates with the loss. GI$_2$ is again the same as the plots in Fig. 2.

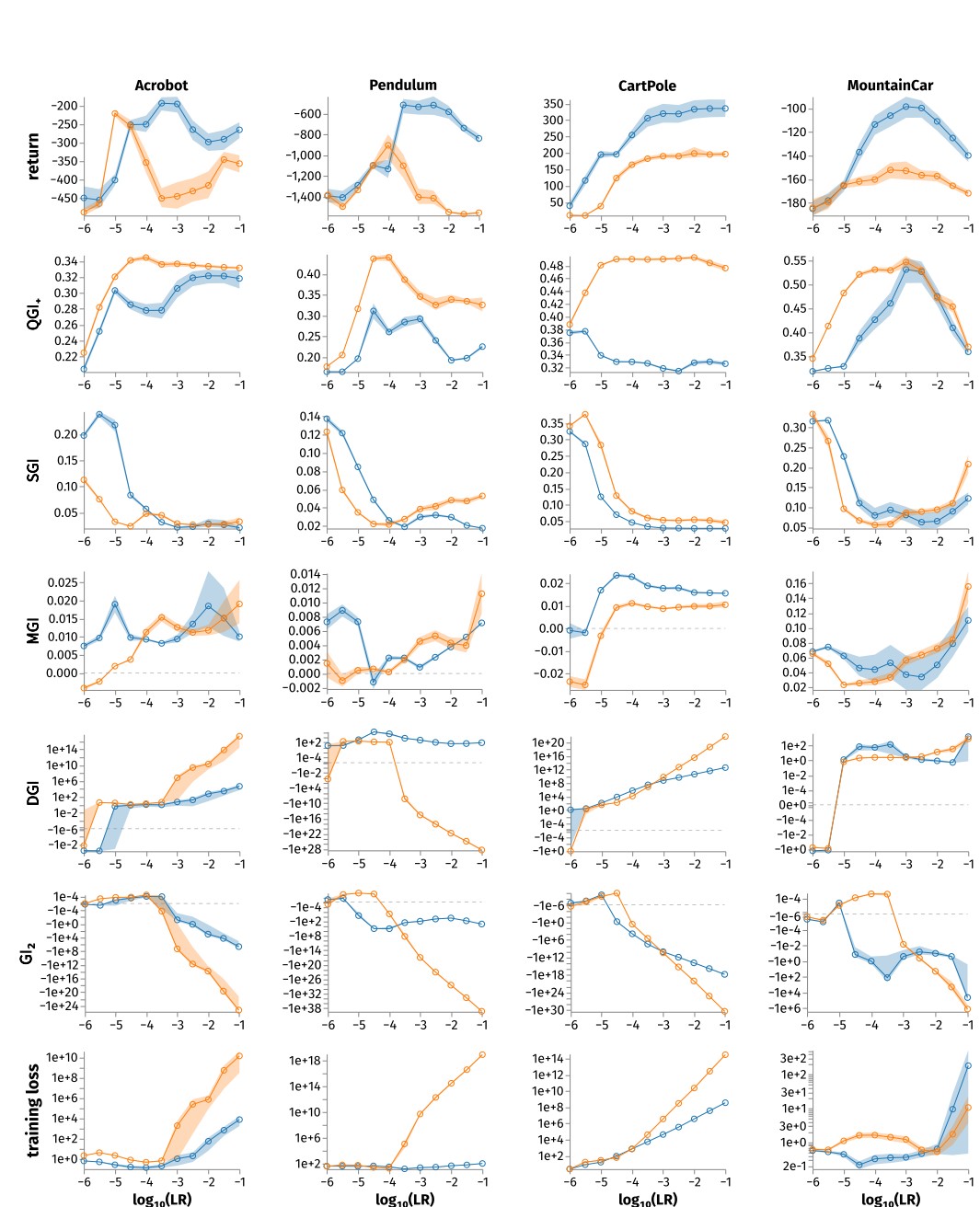

Figure 8: The same as Fig. 7, but using Adam. The results are similar to the SGD case.

# G EXPERIMENTAL DETAILS

Everywhere that we use SGD in this work, we use zero momentum. For Adam (Kingma & Ba, 2014), we use its default hyperparameters.

For Table 1, the 9 learning rates are $\{10^{-1}, 10^{-1.5}, \ldots, 10^{-5}\}$. BN is batch normalization, WN is weight normalization. The "Random" column uses the $Q$-values of the randomly initialized network without any normalization layers. We additionally tested CrossQ (Bhatt et al., 2024) — in our preliminary experiments (not shown), CrossQ achieved around the same returns as LN, though CrossQ requires an additional hyperparameter, the BN momentum. An interesting direction for future work is to study how CrossQ and other normalizations affect the metrics we study.

For all experiments, we use a batch size of 128, a hidden layer width of 128, a discount factor of 0.99, and 200,000 training steps (gradient steps). Environments are: `Pendulum-v1` with a uniformly random initial state-action distribution (Xiao et al., 2021), and an action-discretization from the default continuous space $[-2, 2]$ to instead $\{-2, 0, 2\}$; `MountainCar-v0` with a uniformly random initial state-action distribution; `Acrobot-v1`; and `CartPole-v1`. For any hyperparameters not explicitly specified, we use the PyTorch default. For example, for LN, we use the PyTorch default $\varepsilon = 10^{-5}$.

To compute gradient interference metrics and isometry metrics, we again use a batch of size 128, randomly drawn for each measurement.

For all experiments, to avoid NaN metric values, we clamp the neural network weights after every gradient step to be equal to or under an absolute value of one million. Empirically, this clamping only occurs when using SGD, never when using Adam.

# H LLM USAGE

Following the ICLR author guide, this section describes our LLM usage. We used LLMs to aid and polish our writing. For example, we asked for more concise phrasing, for renaming variables, alternative metric names, and for debugging and improving our LaTeX. We also used LLMs for retrieval and discovery (e.g., finding related work). For instance, we asked LLMs whether our mixed-gradient interference metric appeared in any prior works (in addition to our own, manual searching). We additionally used LLMs to a limited extent for research ideation, such as asking about alternatives to our $\mathsf{ISO}_k$ idea (though ultimately we did not use them). We further used LLMs to help with writing and improving our code.

# I  ALL GRADIENT INTERFERENCE SCATTER PLOTS

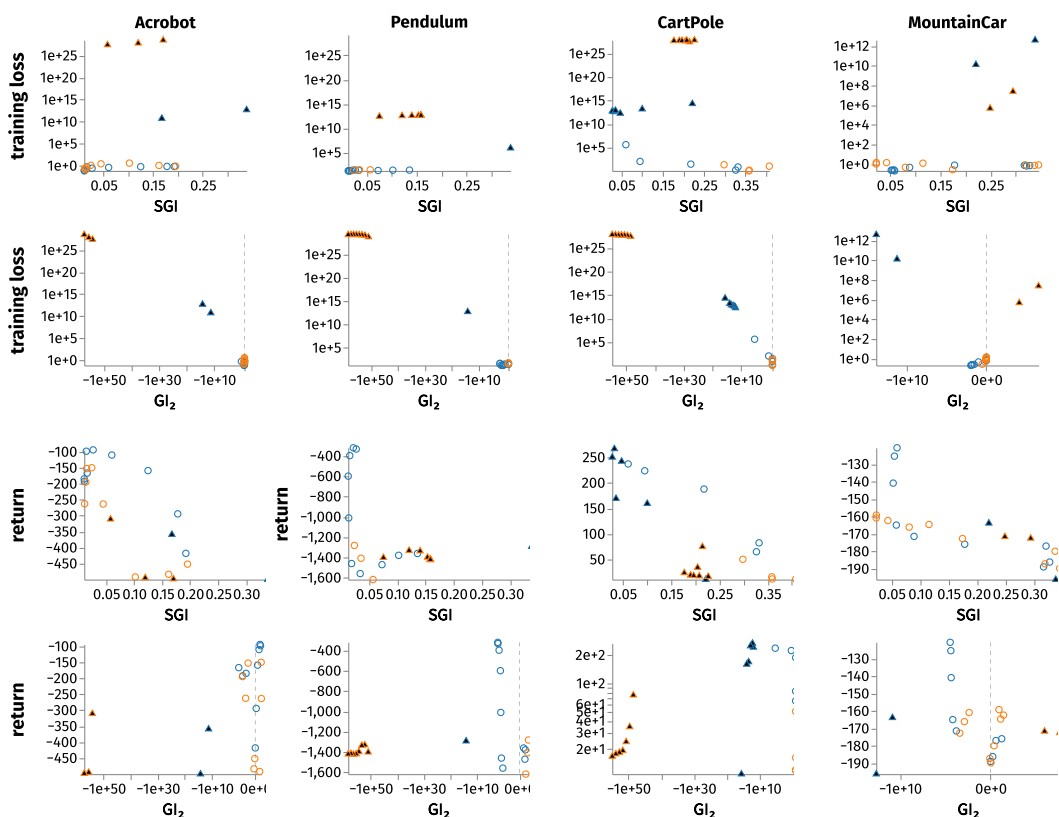

Figure 9: The same as Fig. 1, but with additional plots showing the correlation with return, not only with training loss. On CartPole, SGI correlates negatively with the return, even though it correlates negatively with the loss as well. For Adam (not shown), the correlation between SGI and loss reverses, but the correlation between SGI and return remains negative.

