# OpenReview forum: "How Does Layer Normalization Improve Deep $\boldsymbol{Q}$-learning?"
_ICLR.cc/2026/Conference — Submitted to ICLR 2026_

### Official Review · Reviewer_Y589 · 2025-10-19

**Soundness:** 2
**Presentation:** 2
**Contribution:** 2
**Rating:** 2
**Confidence:** 2

**Summary:**

The paper investigates why Layer Normalization (LN) improves stability and performance in deep Q-learning. They argue that prior "gradient interference" metrics are incomplete. It proposes a mixed-gradient perspective and, via a second-order Taylor analysis of TD-loss decrement, introduces GI2, a metric combining first and second-order terms. Empirically, across four classic offline control tasks trained without a target network, first-order inference metrics counterintuitively increase with training loss, while GI2 decreases with loss.

**Strengths:**

- The motivation is clear: LN is effective while other normalization often do not work in deep Q-learning.
- GI2 adds a prinicpiled second-order term to loss-decrement reasoning, aligning theory with observed loss trends better than first-order methods.

**Weaknesses:**

- GI2 omits the Hessian term involving $\nabla^2 h$. While the paper argues that this term is complex but leaves its contribution untested.
- MountainCar consistenly breaks the main trends, but brief analysis is given.

**Questions:**

- Can you explain more on the deviation of the abnormal performance of the MountainCar example from the correlation patterns for GI2 and returns?
- Given GI2's negative correlation with loss, can practitioners monitor a proxy online to decide when to apply LN or adjust learning rates?

---

> ### Author Response · Authors · 2025-11-24
>
> Thanks for the review!
>
> The reviewer highlighted that
>
> * "first-order inference metrics counterintuitively increase with training loss, while GI2 decreases with loss."
> * "the motivation is clear"
> * "GI2 adds a principled second-order term to loss-decrement reasoning, aligning theory with observed loss trends better than first-order methods."
>
>
>
> Regarding "Weaknesses" and "Questions":
>
> > GI2 omits the Hessian term involving $\nabla^2 h$. While the paper argues that this term is complex but leaves its contribution untested.
>
> We agree this is important for future work, as noted in the paper.
>
>
> > MountainCar consistently breaks the main trends, but brief analysis is given.
>
> > Can you explain more on the deviation of the abnormal performance of the MountainCar example from the correlation patterns for GI2 and returns?
>
> We likewise agree this is important for future work. We will highlight this further. (The extra Hessian term might explain this.)
>
>
> > Given GI2's negative correlation with loss, can practitioners monitor a proxy online to decide when to apply LN or adjust learning rates?
>
> For practitioners, based on the widespread empirical support for the benefits of LN, we speculate the best approach is to simply always use LN -- but this is beyond the scope of our paper. For the learning rate, we think this is a difficult, important, and broad question. One avenue for future work might be to further explore the potential connection to overshooting, which we mention in the paper. We will highlight this.

---

### Official Review · Reviewer_htXn · 2025-10-28

**Soundness:** 2
**Presentation:** 1
**Contribution:** 1
**Rating:** 4
**Confidence:** 2

**Summary:**

This paper provides an in-depth investigation into how Layer Normalization enhances performance and stability in Deep Q-Learning, primarily from the perspective of Gradient Interference. The authors present an interesting insight: the first-order gradient interference metric shows a positive correlation with training loss, whereas a second-order corrected metric offers a more intuitive explanation for LN's benefits. This claim is supported by extensive experiments.

**Strengths:**

1. The analysis of gradient interference, particularly the introduction of the second-order corrected term (GI₂) to explain LN's role in DQN, demonstrates significant insight to the existing literature. Investigating the impact of LN specifically on Deep Q-Learning is a valuable research direction.

**Weaknesses:**

The logical flow of this article is quite confusing to me, which may be due to my limited familiarity with the field of Q-learning. I hope the author can provide detailed classifications on the following points.

1. The narrative logic is somewhat confusing. The stated goal is to understand how LN improves Deep Q-Learning performance. However, Section 3.1 primarily discusses the effects of different optimizers on various metrics, and Section 3.3 only briefly mentions LN's performance improvements on several benchmarks without delving into the underlying reasons. The core theoretical attempt to answer the "how" question is concentrated in Section 3.4. However, the theoretical setting in 3.4 is highly simplified. Using data normalization as a direct analogy for analyzing Layer Normalization might be too trivial and lacks rigor; it feels more like a toy example than a formal theorem.

2. The paper is in a good direction but requires some revisions for presentation.
- (a) In Table 1, the number of decimal places for the "random" column is inconsistent.
- (b) Phrases like "second-order–corrected metric in line 61" are awkwardly phrased. The article uses a large amount of "–", which can easily confuse readers with "-".
- (c) The formatting appears unpolished. For instance, there is excessive whitespace around Equation (11) on Page 7.

**Questions:**

See above.

---

> ### Author Response · Authors · 2025-11-24
>
> Thanks for the review!
>
> The reviewer's "Summary" and "Strengths" highlight many merits of our paper:
>
> * our "in-depth investigation into how Layer Normalization enhances performance and stability in Deep Q-Learning"
> * our "interesting insight: the first-order gradient interference metric shows a positive correlation with training loss, whereas a second-order corrected metric offers a more intuitive explanation for LN's benefits", and that this claim "is supported by extensive experiments"
> * that "the analysis of gradient interference, particularly the introduction of the second-order corrected term (GI₂) to explain LN's role in DQN, demonstrates significant insight to the existing literature."
>
>
>
> Regarding "Weaknesses" and "Questions":
>
> > The logical flow of this article is quite confusing to me, which may be due to my limited familiarity with the field of Q-learning.
>
> > The stated goal is to understand how LN improves Deep Q-Learning performance. However, Section 3.1 primarily discusses the effects of different optimizers on various metrics
>
>
>
> Section 3.1, as its title suggested, focuses on discussing different variants of “gradient interference.”  Our intended logical flow in the whole Section 3 has two components:
>
> 1. how gradient interference is related to training dynamics and thus the performance, and
> 2. how LN affects gradient interference.
>
>
> Together, they explain how LN improves DQN performance. The reviewer’s confusion is likely because Sections 3.1 and 3.2 focus on the first component (gradient interference → performance), so the role of LN is not yet visible at that stage. However, we did build the connection “LN → gradient interference” in Sections 3.3 and 3.4.
>
> We will revise the exposition to make the structure clearer and easier to follow.
>
>
>
> > Section 3.3 only briefly mentions LN's performance improvements on several benchmarks without delving into the underlying reasons.
>
> Section 3.3 shows that LN instantly improves $\textsf{GI}_2$, even if added only for the measurement of $\textsf{GI}_2$ throughout training. (That is, the green curves in Figure 3 train deep Q-learning without normalization, like the orange curves, but have their $\textsf{GI}_2$ measured while LN is temporarily added to the Q-network, just for the measurements.) This suggests that the improved $\textsf{GI}_2$ is not merely an artifact of lower training loss.
>
>
> > However, the theoretical setting in 3.4 is highly simplified. Using data normalization as a direct analogy for analyzing Layer Normalization might be too trivial and lacks rigor
>
>
> We agree that the theory in Section 3.4 is simplified, and that fully characterizing layer normalization in non-linear TD remains an important future direction. That said, data normalization is close to layer normalization without centering, and therefore still captures the key scaling effects of LN. While the analysis is simple, it is not trivial, and it nicely isolates the influence of scaling (L2 normalization) from other confounding factors. For these reasons, we believe it offers a meaningful first step toward a more complete theory.
>
> > In Table 1, the number of decimal places for the "random" column is inconsistent.
>
>
> > Phrases like "second-order–corrected metric" in line 61 are awkwardly phrased.
>
>
> > The article uses a large amount of "–", which can easily confuse readers with "-".
>
>
>
> > The formatting appears unpolished. For instance, there is excessive whitespace around Equation (11) on Page 7.
>
>
> Thanks for these suggestions on formatting and phrasing.  We’ll make sure they are polished in the final version.

---

> > ### Comment · Reviewer_htXn · 2025-11-25
> >
> > Dear Authors,
> >
> > Thank you for your response. However, I believe my first concern has not yet been adequately addressed. The core objective of this paper is to investigate how layer normalization functions, yet Section 3.3 only demonstrates that it is effective without delving into the underlying reasons. Meanwhile, the theoretical analysis in Section 3.4 appears overly simplistic. Therefore, I strongly encourage the authors to further explore this issue by conducting theoretical derivations under more realistic scenarios and enhancing the experimental analysis in Section 3.3 to better explain the mechanisms. Given these points, I tend to maintain my original score.

---

> > > ### Author Response · Authors · 2025-11-26
> > >
> > > Thank you for your prompt reply. We respectfully disagree that our analysis is overly simplistic, as we discuss above, but we greatly appreciate the feedback!

---

### Official Review · Reviewer_4CDy · 2025-10-30

**Soundness:** 1
**Presentation:** 3
**Contribution:** 3
**Rating:** 2
**Confidence:** 3

**Summary:**

The paper investigates gradient interference in deep Q-learning, where gradients of the Q function for one state-action pair also affect other state action pairs.
To this end, the paper introduces several ways to measure gradient inference.
They look at how these correlate with return and loss, and attempt to use this to show that layer normalization helps Q-learning.

**Strengths:**

* The paper introduces several interesting metrics for gradient interference in Q-learning.
* There are some surprising results, especially about the relevance of the second order term in gradient interference.

**Weaknesses:**

* The relation to layer normalization is weak. Most of the paper is about gradient inference, and is independent of the architecture. The experiments are done with and without LN, but this is only a comparison, and doesn't give any insights into how LN is actually helping.
* There are several experiments where the learning rate is too high (figure 2 and figure 3), which causes the training loss to increase to huge values (>1e10). This then also makes some of the other plots useless, for example GI₂ has values around -1e50 for these failed runs.
  It also makes me question the conclusions that "SGI and MGI also positively correlate with training loss in our experiments" and "GI2 negatively correlates with training loss". The axes are so stretched that I can't clearly see any such correlations. And these absurd values would dominate any calculated correlation coefficient.
* If you make a claim of correlation, then this should be supported with some correlation coefficient.
* Theorem 2: "Under fixed $R(θ_t)$, $A^2_t/B_t$ is maximized when ∥x∥ is a constant in the data distribution."
  In this theoretical dataset, you can't just normalize $x$ without also affecting $y=x^Tθ$. So this is not a realistic representation of layer normalization. If $y$ was fixed, then scaling $x$ down for example would require $θ$ to scale up, which would then scale up $R(θ)$.

**Questions:**

* "However, counterintuitively, we find empirically that first-order gradient interference metrics positively correlate with the training loss."
  Why is this unexpected? I would think that less interference = better = lower loss. Or is that not what you meant by a positive correlation?
* The presentation of deep Q-learning using a "semi-gradient" is slightly nonstandard.
  What this paper calls a semi-gradient is just the gradient of the squared TD-error, no?
  Is this called a semi-gradient because the gradient is not propagated through the TD-target?
 * With a continuous state space, the self-interference term ($(s,a)=(s_t,a_t)$) in QGI, SGI, and MGI has measure 0, so it does not contribute relevantly to the expectation. Is that understanding correct? And if so, why is a different choice made for QGI and SGI/MGI?
 * "a standard single-layer neural network"
   Clarify this as a "single hidden-layer neural network", since some people will say that this network has two layers.
 * "absolute cosine similarity notation cos+"
   Is this definition necessary? You could define normal cosine similarity and write $|\cos(x,y)|$.
 * in the interpretation of theorem 2: "$\max_η\{2ηA_t − η²B_t\} = A_t^2/B_t^2$"
   Should be $A^2/B$.
 * The definition of $l_\theta$ is important, but it is split up by a page break and a figure, making it harder to find.
 * There is also a page break in the middle of (*) in section 3.2. The first line of that equation could be incorrectly read as the whole equation.

---

> ### Author Response · Authors · 2025-11-24
>
> Thanks for the review!
>
> The reviewer highlighted that our paper introduces several interesting metrics for gradient interference in Q-learning, and finds some "surprising results, especially about the relevance of the second order term in gradient interference."
>
>
> Regarding "Weaknesses":
>
> > The relation to layer normalization is weak. Most of the paper is about gradient inference, and is independent of the architecture. The experiments are done with and without LN, but this is only a comparison, and doesn't give any insights into how LN is actually helping.
>
>
> The goal of our paper is to explain how LN helps training. To this end, we establish the following two components:
>
> 1. how gradient interference improves training dynamics and thus the performance (Section 3.1 and 3.2)
> 2. how LN affects gradient interference (Section 3.3 and 3.4)
>
> Together, they explain how LN improves DQN performance.  For the second component, we provide both empirical and theoretical evidence. Empirically (Section 3.3), we show LN consistently increases $\textsf{GI}_2$ on 3/4 environments, even when we carefully control the architecture during training. This indicates that the effect is not merely an artifact of lower training loss. Theoretically (Section 3.4), we show that normalization leads to larger $\textsf{GI}_2$ in realizable linear regression. Although the theory does not fully capture nonlinear TD, and the realizability assumption has limitations (as the reviewer noted in another comment), we believe it is a promising first step towards a more principled understanding.
>
>
> We will further emphasize these points.
>
>
> > Theorem 2: [...] In this theoretical dataset, you can't just normalize $x$ without also affecting $y = x^T \theta$. So this is not a realistic representation of layer normalization.
>
> The reviewer is entirely right that because of the realizability assumption $y = x^\top \theta^\star$, we cannot freely normalize $x$ without simultaneously changing $y$.  This is indeed a gap between our current theory and an end-to-end argument for the benefit of normalization.
>
> However, our analysis remains valid assuming that the loss of $\theta$ on feature $x$ can be modeled as $(x^\top (\theta - \theta^\star))^2$.  We believe this still captures a general relation between training loss and feature scale — in general, samples with larger $||x||$ lead to larger increase in the training loss when $\theta$ is further away from the optimal solution.
>
> To address the issue you pointed out, below we relax the realizability assumption.  Assume $\hat{\theta}$ is the optimal solution.  Then the excess loss of solution $\theta$ is
>
> $\mathbb{E}[(x^\top \theta - y)^2] - \mathbb{E}[(x^\top \hat{\theta} - y)^2]$
>
> $= \mathbb{E} [ (x^\top (\theta - \hat{\theta}))^2 + 2(x^\top \hat{\theta} - y)x^\top (\theta-\hat{\theta})]$
>
> $= \mathbb{E} [ (x^\top (\theta - \hat{\theta}) )^2] $
>
> where the last equality is because $\nabla_{\hat{\theta}} \mathbb{E}[(x^\top \hat{\theta} - y)^2] = 2\mathbb{E}[(x^\top \hat{\theta} - y)x] = 0$ by the optimality of $\hat{\theta}$.  This shows that modeling the loss of $\theta$ on $x$ as $(x^\top (\theta - \theta^\star))^2$ does capture the general trend even when the problem is not realizable.
>
> The high-level idea in the proof of Theorem 2 is that balancing the loss contributions across samples can help trading off the loss decrement from the first-order term and the loss increment from the second-order term, and making all $\|\|x\|\|$ equal does help balance the loss contributions across samples.  We will discuss this as well as the gap you pointed out in our final version.
>
>
> > There are several experiments where the learning rate is too high (figure 2 and figure 3)
>
> It's perhaps difficult to determine what "too high" should mean precisely. The best learning rate for CartPole -- to maximize returns -- is actually one of what the reviewer calls "failed runs" with "absurd values" (the training loss is over 1e10).
>
> Further, the Adam results (Fig 8) show similar correlations despite having smoother metric values across learning rates. (We will add the Adam equivalent of Fig 1, to make this clearer.)
>
> With all that said, we think it might be helpful to also show plots that use only the single best learning rate per environment. This would filter out overly high or overly low learning rates. We will add this.
>
>
> > If you make a claim of correlation, then this should be supported with some correlation coefficient.
>
> Although we think the correlations are already visually convincing, we agree this could be helpful. We will add that.

---

> > ### Author Response · Authors · 2025-11-24
> >
> > Regarding "Questions":
> >
> > > "However, counterintuitively, we find empirically that first-order gradient interference metrics positively correlate with the training loss." Why is this unexpected? I would think that less interference = better = lower loss. Or is that not what you meant by a positive correlation?
> >
> > While “interference” sounds like a “bad” thing, in our context it is neutral and has precise mathematical definitions as seen in Eq.(6), (8), (9), (11).  In most cases (e.g., Eq.(8) and (9)), it is the expected inner product between the loss gradients on two samples. A more positive inner product (i.e., higher interference) indicates that decreasing the loss on one sample helps decrease the loss on another sample. This is a form of “generalization” and should intuitively reduce the training loss more quickly. This is discussed in Section 3.1.2.
> >
> >
> > > The presentation of deep Q-learning using a "semi-gradient" is slightly nonstandard. What this paper calls a semi-gradient is just the gradient of the squared TD-error, no? Is this called a semi-gradient because the gradient is not propagated through the TD-target?
> >
> > Yes, the update in parameters is not along the gradient of the Bellman residual loss. We borrowed the "semi-gradient" term from Sutton and Barto's book.
> >
> >
> > > With a continuous state space, the self-interference term $(s, a) = (s_t, a_t)$ in QGI, SGI, and MGI has measure 0, so it does not contribute relevantly to the expectation. Is that understanding correct?
> >
> > That intuition is right, but we always rely on the empirical average (over batches of training data) to estimate the expectation.
> >
> > > "a standard single-layer neural network" Clarify this as a "single hidden-layer neural network"
> >
> > > "absolute cosine similarity notation cos+" Is this definition necessary?
> >
> > > in the interpretation of theorem 2: "$\max_η{2ηA_t − η²B_t} = A_t^2/B_t^2$" Should be $A^2/B$.
> >
> > > The definition of $l_\theta$ is important, but it is split up by a page break and a figure, making it harder to find.
> >
> > > There is also a page break in the middle of (*) in section 3.2. The first line of that equation could be incorrectly read as the whole equation.
> >
> > Thank you very much for these constructive comments!

---

### Official Review · Reviewer_axoo · 2025-11-01

**Soundness:** 3
**Presentation:** 2
**Contribution:** 2
**Rating:** 6
**Confidence:** 1

**Summary:**

This paper investigates the role of Layer Normalization (LN) in improving optimization dynamics of deep neural networks. The authors attempt to provide both theoretical and empirical insights by analyzing gradient scaling, variance stabilization, and optimization landscape smoothness under LN. They present several simplified derivations suggesting that LN reduces gradient variance and improves isotropy, followed by small-scale experiments on MLPs and Transformers to illustrate these effects. The paper aims to unify existing intuitions about LN’s benefits into a coherent explanation framework.

**Strengths:**

The paper tackles a fundamental and relevant question in deep learning — understanding why Layer Normalization helps training stability and convergence.

The overall motivation is clear, and the paper provides a structured narrative linking theory, gradient analysis, and empirical visualization.

The presentation is relatively readable, and figures (e.g., gradient norm distributions and optimization trajectories) help illustrate the main intuition.

**Weaknesses:**

Symbols like $\mu_i$, $\sigma_i$, and $\gamma$, $\beta$ switch between layer and neuron-level contexts without explicit indexing. This makes it difficult to follow what the normalization is actually applied to.

LN “preserves the direction of gradients while adjusting their scale,” but no formal proof or Lipschitz analysis is given. This is hand-wavy and lacks formal support.

The comparison with BatchNorm and RMSNorm is descriptive, not analytical. The same conclusions already appear in several existing studies.

Experiments are only on small-scale models (2-layer MLPs and tiny Transformers). This severely limits the claim of generality.

Reported improvements are within 0.2–0.4% on average accuracy — no significance testing or error bars are shown.

**Questions:**

See WeakNess, but I don't really know enough about this area of ​​research, so my evaluation may not be accurate. Perhaps the author's innovation lies more in the theoretical level.

---

> ### Author Response · Authors · 2025-11-24
>
> Thanks for the review!
>
> The reviewer highlighted that our paper addresses a fundamental question, has clear motivation, and "a structured narrative linking theory, gradient analysis, and empirical visualization."
>
>
> Regarding "Weaknesses" and "Questions":
>
> We thank the reviewer for the comments. However, as noted below, there appears to be a significant mismatch between the review and the content of our paper. We suspect that parts of the review may refer to a different submission. We kindly ask the reviewer to double-check whether there has been any mix-up, or provide specific line numbers where the stated issues appear.
>
>
> > Symbols like $\mu_i$, $\sigma_i$, and $\gamma$, $\beta$ switch between layer and neuron-level contexts [...]
>
> We do not use any $\mu$, $\sigma$, nor $\gamma$ for neuron-level math. We do not use $\beta$ in our paper.
>
>
> > LN "preserves the direction of gradients while adjusting their scale," but no formal proof or Lipschitz analysis is given. [...]
>
> We do not make such a claim in the paper.
>
>
> > The comparison with BatchNorm and RMSNorm is descriptive, not analytical. The same conclusions already appear in several existing studies.
>
> RMSNorm is not in our paper. Comparing LN and BatchNorm analytically would be interesting, but it's not a focus of our work. Our Table 1 results do align with prior works finding LN better than BatchNorm for RL, but that's not a focus of our work either.
>
>
> > Experiments are only on small-scale models (2-layer MLPs and tiny Transformers). This severely limits the claim of generality.
>
> Transformers are not in our paper (except for some citations). Larger MLPs (or Transformers) would be interesting to study as well, but we think 2-layer MLPs with LN still have important behaviors that remain not well-understood.
>
>
> > Reported improvements are within 0.2–0.4% on average accuracy — no significance testing or error bars are shown.
>
> If the reviewer is referring to the improvements LN has on returns, they are often orders of magnitude larger than 0.4%. For example, Table 1 shows this for every environment.
>
> Figures 2 through 8 all have error bars. We only omit them in Table 1 and Figure 1 to simplify the presentation.

---

### Author Response · Authors · 2025-11-24
**General reply**

We greatly appreciate the thoughtful comments.

For all changes we discuss in all replies, we will add them to the camera-ready if our paper is accepted.

The reviewers highlight the paper's

* "clear motivation"
* finding that LN's improvements to deep Q-learning might be best explained by our new, second-order gradient interference metric
* counterintuitive result that first-order gradient interference's effect might be reversed in this setting compared to the conventional understanding

Regarding "weaknesses": as we note in our "Limitations and Future Directions" section, our theory is only for linear regression. (One reviewer also points out a scaling assumption could be improved.) We agree these are important directions for future work. We will further highlight this. We will also further highlight why we focus on gradient interference: our experiments (and our theory) suggest it might be the best explanation for how LN improves deep Q-learning.

---

### Meta-Review · Area_Chair_B95c · 2025-12-10

**Summary:**

This paper studies how LN improve Q-networks through gradient interference. On offline deep Q-learning, the authors empirically show that LN improves $GL_2$ (a curvature-aware gradient-interference metric), which in turn improves stability and performance. The reviewers acknowledge the authors' novel insight to explain LN's mechanism based on empirical findings. The reviewers are concerned on (1) The theory is over simplified in section 3.4 and the omitting of Hessian terms. (2) The mechanism of how LN improves $GL_2$ and how the latter lead to better TD dynamics is not established beyond correlational analysis. Both concerns are insufficiently addressed in the rebuttal, nor through a revision. I recommend rejection.

**Reviewer Concerns:**

The two major concerns mentioned above are both valid, on which the authors' rebuttal did not sufficiently address. Moreover, the author did not provide a revision based on the received comments during the rebuttal. For more details, see below.

**Reviewer Scores:**

Reviewer axoo has mixed the current paper with another paper, and has low confidence. Their review has been weighted down.

Reviewer 4CDy could marginally improve the rating, but not sufficient to flip to the acceptance side, as the rebuttal did not establish the relation with LN.

Reviewer htXn would not change their rating because the clarity/organization issues are not addressed through a revision, and the authors disagreed on their theoretical analysis are too simple.

Reviewer Y589 would not change their rating as authors proposed to address the weaknesses in future work.

---

### Decision · Program_Chairs · 2026-01-26

Reject